# Emergence of Hyper-Epidemic Clones of Enterobacterales Clinical Isolates Co-Producing KPC and Metallo-Beta-Lactamases during the COVID-19 Pandemic

**DOI:** 10.3390/pathogens12030479

**Published:** 2023-03-18

**Authors:** Diego Faccone, Sonia A. Gomez, Juan Manuel de Mendieta, María Belén Sanz, Mariano Echegorry, Ezequiel Albornoz, Celeste Lucero, Paola Ceriana, Alejandra Menocal, Florencia Martino, Denise De Belder, Alejandra Corso, Fernando Pasterán

**Affiliations:** 1Servicio Antimicrobianos, National Reference Laboratory in Antimicrobial Resistance (NRLAR), National Institute of Infectious Diseases (INEI), ANLIS “Dr. Carlos G. Malbrán”, Ave. Velez Sarsfield, 563, Buenos Aires City 1281, Argentina; dfaccone@anlis.gob.ar (D.F.); sgomez@anlis.gob.ar (S.A.G.); ; acorso@anlis.gob.ar (A.C.); 2Consejo Nacional de Investigaciones Científicas y Técnicas (CONICET), Godoy Cruz, Buenos Aires City 2290 (C1425FQB), Argentina

**Keywords:** Carbapenemase, Enterobacterales, ST307, COVID-19

## Abstract

Background. The global spread of carbapenemase-producing Enterobacterales has become an epidemiological risk for healthcare systems by limiting available antimicrobial treatments. The COVID-19 pandemic worsened this scenario, prompting the emergence of extremely resistant microorganisms. Methods. Between March 2020 and September 2021, the NRL confirmed 82 clinical Enterobacterales isolates harboring a combination of *bla*_KPC_ and MBL genes. Molecular typing was analyzed by PFGE and MLST. Modified double-disk synergy (MDDS) tests were used for phenotypic studies. Results. Isolates were submitted from 28 hospitals located in seven provinces and Buenos Aires City, including 77 *K. pneumoniae*, 2 *K. oxytoca*, 2 *C. freundii,* and 1 *E. coli*. Almost half of *K. pneumoniae* isolates (n = 38; 49.4%), detected in 15 hospitals, belong to the CC307 clone. CC11 was the second clone, including 29 (37.7%) isolates (22, ST11 and 7, ST258) from five cities and 12 hospitals. Three isolates belonging to CC45 were also detected. The carbapenemase combinations observed were as follows: 55% *bla*_KPC-2_ plus *bla*_NDM-5_; 32.5% *bla*_KPC-2_ plus *bla*_NDM-1;_ 5% *bla*_KPC-3_ plus *bla*_NDM-1;_ 5% *bla*_KPC-2_ plus *bla*_IMP-8_; and 2.5% strain with *bla*_KPC-2_ plus *bla*_NDM-5_ plus *bla*_OXA-163_. Aztreonam/avibactam and aztreonam/relebactam were the most active combinations (100% and 91% susceptible, respectively), followed by fosfomycin (89%) and tigecycline (84%). Conclusions. The MDDS tests using ceftazidime-avibactam/EDTA and aztreonam/boronic acid disks improved phenotypic classification as dual producers. The successful high-risk clones of *K. pneumoniae*, such as hyper-epidemic CC307 and CC11 clones, drove the dissemination of double carbapenemase-producing isolates during the COVID-19 pandemic.

## 1. Introduction

The emergence and global spread of carbapenemase-producing Enterobacterales (CPE) have become an epidemiological risk for healthcare systems and a serious threat to antimicrobial treatment [1]. Currently, *Klebsiella pneumoniae* carbapenemase (KPC), OXA-48-like, New Delhi Metallo-beta-lactamase (NDM), Verona Integron-Encoded Metallo-beta-lactamase (VIM), and Imipenemase (IMP) are the most frequently detected carbapenemases worldwide [2]. The major burden of CPE infections is due to the global spread of successful *K. pneumoniae* clones, especially clonal complex (CC)11 [3,4]. Moreover, *K. pneumoniae* belonging to sequence type (ST) 25, ST307, and ST147 were recently added to the large family of successful clones in this species [4].

Since the first detection of Enterobacterales-producing KPC-2 or NDM-1 in Argentina in 2006 and 2014, respectively [5,6], both carbapenemases were able to acquire endemic proportions even before the COVID-19 pandemic. The dissemination of KPC in Argentina was mainly driven by *K. pneumoniae* ST258 [7] and more recently by other minor clones such as ST25 [8]. In contrast, the local dissemination of NDM was driven by multiple Gram-negative species and clones [9].

During the COVID-19 pandemic, the emergence of extremely resistant microorganisms and an increase in the incidence of resistance to carbapenems were documented, possibly, due to the increased use of broad-spectrum antibiotics [10,11]. During the first wave of COVID-19 (2020) in Argentina, carbapenemase-producing *K. pneumoniae* increased from 20% (2019) to 30% in prevalence according to the National Surveillance System WHONET-Argentina (http://antimicrobianos.com.ar/ATB/wp-content/uploads/2022/11/Vigilancia-Nacional-de-la-Resistencia-a-los-Antimicrobianos-Red-WHONET-Argentina-Tendencia-2010-2021.pdf) (Accessed on 16 March 2023). In parallel, since May 2020, an increasing number of isolates with co-production of carbapenemases were confirmed by the NRL. Because of this emergence, the NRL issued a national alert [12]. Subsequently, a regional alert issued by the Pan-American Health Organization-World Health Organization reported the emergence of Enterobacterales’ co-producing of KPC plus NDM during the COVID-19 pandemic by the NRLs from Uruguay, Ecuador, and Guatemala, together with Argentina, among other countries [13,14]. In this study, we characterize 82 Enterobacterales clinical isolates with the emergent combination of KPC plus a metallo-β-lactamase (MBL).

## 2. Materials and Methods

### 2.1. Bacterial Isolates

Between March 2020 and September 2021, a total of 699 Enterobacterales clinical isolates suspicious of carbapenemase production were submitted to the NRL for molecular characterization [15]. A total of 580 isolates were positive for at least one carbapenemase gene, with 118 isolates being positive for the combination of *bla*_KPC_ and an MBL gene. A sample of 82/118 representative isolates positive for *bla*_KPC_ and an MBL gene were selected for further molecular study, considering one isolate per patient referred from all geographic regions, institutions, and years of isolation and excluding isolates that may have been part of an outbreak. Species identification was performed by MALDITOF (Bruker Daltonics Billerica, Massachusetts, MA, USA).

### 2.2. Molecular Methods

Carbapenemase confirmation was performed using an *in house* multiplex PCR to detect *bla*_KPC_, *bla*_NDM_, *bla*_IMP_, *bla*_VIM_, and *bla*_OXA-48-like_ genes. The reaction was set up with 200 µM of dNTPs, 1.5 mM of MgCl_2_, 1X buffer, and 2U of Taq polymerase (Invitrogen) at a final volume of 50 µL. The reaction primer concentration was 200 nM for VIM, NDM, and OXA-48-like, 400 nM for KPC, and 480 nM for IMP (Appendix A). A triplex PCR to detect *bla*_CTX-M_ and *bla*_PER_ ESBL-coding genes plus *bla*_CMY_ plasmid-borne AmpC was additionally used. The reaction was set up with 200 µM of dNTPs, 1.5 mM of MgCl_2_, 1X buffer, and 1U of Taq polymerase (Invitrogen) at a final volume of 25 µL. The primer concentration was 200 nM for PER and CMY and 600 nM for CTX-M (Appendix A). Amplification products were separated by electrophoresis in a 1% agarose gel and visualized with ethidium bromide-staining. Sanger sequencing (ABI PRISM 3100 o 3730, Applied Biosystems) was used to confirm gene variants. To discriminate between Tn*4401*-related or ΔTn*3*-related *bla*_KPC_-containing genetic platforms, we performed PCR combining specific primers as previously demonstrated [16].

The genetic relatedness among the isolates was evaluated by XbaI-digested pulsed-field gel electrophoresis (PFGE) using a CHEF-DR^®^ III System (Bio-RadTM, Hercules, CA, USA) and interpreted as previously reported [16]. DNA fragments were resolved in 1% agarose gel by applying a switch time of 2.2 to 54.2 s for 20 h at 14 °C. At least one isolate of each PFGE profile was selected for multi-locus sequence typing (MLST).

### 2.3. Phenotypic and Microbiological Tests

Isolates were phenotypically screened by a synergy disks test using 10 µg-carbapenem and 30 µg-aztreonam disks placed ca. 20 mm and 10 mm, respectively, to a disk containing 300 µg-amino-phenyl boronic acid (APB). Additionally, 10 µg-carbapenem and 10/4 µg-ceftazidime/avibactam disks were placed to ca. 20 mm and 10 mm, respectively, to a disk containing 750 µg-EDTA [12,17]. The Carbapenem Inactivation Test (CIM) plus the EDTA-based CIM and Carba NP test were used as additional confirmatory tests, following protocol and interpretation criteria by CLSI [18]. *Escherichia coli* ATCC 25922, *K. pneumoniae* ATCC 700,603, and *K. pneumoniae* ATCC BAA-1705 were used as quality control strains across all tests.

### 2.4. Susceptibility Testing

Susceptibility to relevant antimicrobials was determined by disk diffusion following CLSI guidelines [18]. Susceptibility to aztreonam plus avibactam and aztreonam plus relebactam were determined by agar dilution, using a fixed 4 µg/mL of each inhibitor. As no breakpoints are yet available for these combinations, we used the aztreonam alone breakpoint defined by EUCAST (susceptible ≤ 1 mg/L). Colistin (COL) susceptibility was evaluated by disk elution and agar spot or drop tests [18,19]. The fosfomycin and tigecycline breakpoints were those defined by EUCAST and FDA, respectively [20,21].

## 3. Results

### 3.1. Epidemiological Analysis of the Isolates

The 82 isolates co-producing *bla*_KPC_ plus an MBL gene included in this study are described in detail in Table 1. The selected isolates were submitted from 28 hospitals located in seven provinces or Buenos Aires City (CABA). Most isolates (58/82; 70.1%) were submitted from 16 hospitals from CABA, 2 of these hospitals referred 24 (H1) and 10 (H2) isolates, respectively. A total of 17 isolates were submitted from six hospitals in Buenos Aires Province, and one institution referred 11/17 isolates. Santa Fe province submitted two isolates while the provinces of Chaco, Corrientes, Entre Ríos, Neuquén, and Tucuman submitted only one isolate each as detailed in Table 1.

Four Enterobacterales species were identified among the 82 isolates with *K. pneumoniae* (77/82) being the most common one, followed by *K. oxytoca* (2/82), *C. freundii* (2/82),- and *E. coli* (1/82) (Table 1). The samples were obtained mainly from blood (21/82), urine (20/82), rectal swabs (16/82), and tracheal aspirate (8/82). According to the available data, 77% (63/82) of the isolates were recovered from males. The age of the patients ranged between 3 and 88 years, with an average of 52 years old (n = 55).

### 3.2. Molecular Characterization and Genetic Relatedness of Isolates

The 82 isolates harbored the *bla*_KPC_ gene, 77 co-harbored *bla*_NDM,_ and 4 isolates co-harbored *bla*_IMP_. One *K. pneumoniae* isolate co-produced *bla*_KPC,_
*bla*_NDM,_ and *bla*_OXA-48-like_ carbapenemase genes (Table 1 and Appendix A).

Genetic relatedness was analyzed by PFGE among the 77 *K. pneumoniae* isolates, grouping them into 36 profiles (Table 1 and Appendix A). Pulsotype A grouped 22 isolates (28.6%) submitted from five hospitals from CABA (n = 3), Buenos Aires (n = 1), and Santa Fe (n = 1) (Table 1 and Appendix A). All nine isolates grouped in pulsotype R and three in pulsotype D were submitted from a single hospital (H1) from CABA. Pulsotypes B and C, with four and three isolates each, were all recovered from a single hospital in CABA (H2). The remaining 31 pulsotypes were represented by one or two isolates each (Table 1 and Appendix A). Finally, five additional non-*K. pneumoniae* isolates were submitted from five institutions: two genetically unrelated *K. oxytoca*, two genetically unrelated *C. freundii,* and one *E. coli* (ST4774).

MLST was determined in a selection of 40 *K. pneumoniae* isolates, including at least one of each pulsotype defined by PFGE (Table 1 and Appendix A). All isolates grouped in a particular pulsotype were inferred to belong to the same ST. The most common clone was CC307, including 38 (49.4%) *K. pneumoniae* isolates, detected in 15 out of 27 hospitals and comprising 37 *K. pneumoniae* ST307 and 1 ST5993 (SLV307). The second *K. pneumoniae* clone was CC11, including 29 (37.7%) isolates from five cities and 12 hospitals. Of these, 22 isolates belonged to ST11 and 7 isolates to ST258 (Table 1 and Appendix A). The third clone in abundance was represented by three isolates (3.9%) obtained in the same institution and belonged to CC45 (ST45). The details of the remaining seven STs (9%) can be seen in Table 1, all represented by only one isolate each. Comparing the infection site among the clones, 4/7 of ST258 were recovered from blood, while ST307 and ST11 were the only *K. pneumoniae* isolates recovered from respiratory tract samples.

The carbapenemase allelic variants were determined in the same 40 selected *K. pneumoniae* isolates for MLST, including at least one of each pulsotype defined by PFGE (Table 1 and Appendix A). The combinations observed were as follows: 22 (55%) isolates harbored *bla*_KPC-2_ plus *bla*_NDM-5_, 13 (32.5%) isolates *bla*_KPC-2_ plus *bla*_NDM-1,_ 2 (5%) isolates with *bla*_KPC-3_ plus *bla*_NDM-1_, 2 (5%) isolates with *bla*_KPC-2_ plus *bla*_IMP-8_, and 1 (2.5%) isolate with *bla*_KPC-2_ plus *bla*_NDM-5_ plus *bla*_OXA-163_. Analyzing the non-*K. pneumoniae* isolates, we found that both *C. freundii* and *E. coli* isolates were confirmed to harbor *bla*_KPC-2_ plus *bla*_NDM-1_. One *K. oxytoca* co-harbored *bla*_KPC-2_ plus *bla*_NDM-5_ while the other one co-harbored *bla*_KPC-2_ plus *bla*_IMP-8_.

Analysis of the close genetic environment of *bla*_KPC_ among the selected 40 *K. pneumoniae* isolates revealed that 57.5% (23/40) harbored *bla*_KPC_ in the ΔTn*3* variant 1a/b, already described in Argentina (Table 1) [16]. Surprisingly, twenty isolates, harbored *bla*_KPC_ in IS*Kpn8*-ΔTn*3*–*bla*_KPC-2_-IS*Kpn6-like* and only three in IS*Kpn8*-ΔTn*3*–Δ*bla*_TEM-1_-*bla*_KPC-2_-IS*Kpn6-like,* known as Variant 1a and which, to the best of our knowledge, used to be the most abundant genetic environment reported in Argentina in non-*K. pneumoniae* species [7,16]. The remaining isolates, 17/40 (42.5%), harbored *bla*_KPC_ inserted in the typical Tn*4401*a, including both *bla*_KPC-3_ producers. The major clones, ST307 and ST11, showed no association with any particular *bla*_KPC_ genetic platform. Instead, *bla*_KPC-2_ in ST258 was found only in the Tn*4401a* element as previously described [7].

Plasmid-borne ESBLs (*bla*_CTXM_ and *bla*_PER_) and AmpC (*bla*_CMY_) genes were screened among all isolates (Appendix A). *bla*_CTX-M_ was detected in most *K. pneumoniae* isolates (63/77; 81.8%) and one *K. oxytoca*. Additionally, *bla*_CMY_ was detected in seven *K. pneumoniae* (9.1%), four of these co-harbored *bla*_CTX-M_. *bla*_CMY_ was also detected in one non-*K. pneumoniae* isolate (*E. coli*). *bla*_PER-2_ was not detected at all.

### 3.3. Susceptibility Testing of Double Carbapenemase Producers

Susceptibility testing showed that all the isolates were resistant to cephalosporins, monobactams, carbapenems, ceftazidime/avibactam, and imipenem/relebactam, but remained susceptible to fosfomycin (89%), tigecycline (84%), colistin (60%), minocycline (49%), amikacin or gentamicin (12%), ciprofloxacin (4%), and trimethoprim-sulfamethoxazole (1%). Aztreonam/avibactam MIC_50_ and MIC_90_ values were 0.12 µg/mL and 0.25 µg/mL, respectively (Table 2). No strain had a MIC >1 µg/mL, the non-susceptibility breakpoint for aztreonam alone according to EUCAST. Aztreonam/relebactam was less active than aztreonam/avibactam, with MIC_50_ and MIC_90_ of 0.5 µg/mL and 2 µg/mL, respectively (Table 2). The activity of aztreonam/avibactam was similar among the different *K. pneumoniae* STs (MIC differences ≤ 2 log_2_). In contrast, a significant association was found between aztreonam/relebactam activity and *K. pneumoniae* STs, with MIC_50_/MIC_90_ >1 µg/mL for ST258 only (Table 2). When comparing groups of isolates according to the carbapenemases content, no differences were observed (Table 2).

The resistance profile for almost all remaining antimicrobials was similar among STs. Exceptions were observed with ST307 which was more resistant to m inocycline (73% vs. 18% of resistance) and tigecycline (27% vs. 5% of resistance) than ST11 but remained more susceptible to colistin (24% vs. 59% of resistance).

### 3.4. Performance of Phenotypic Detection Tests of Double Carbapenemase Producers

All 82 isolates were confirmed as carbapenemase-producers by the Carba NP test. The mCIM/eCIM tests suggested the expression of a serine enzyme in 91% of the strains, while in the remaining 9%, the results were compatible with the expression of an MBL. APB and EDTA synergy tests, placing inhibitor-containing disks close to carbapenems, failed to classify the simultaneous production of Class A (KPC) plus Class B (MBL) carbapenemases in 66/82 (80.5%) of the cases. The errors in classification were as follows: 36/82 (43.9%) of the isolates were misclassified as single carbapenemase producers (21% as Class A or 23% as Class B producers), while the remaining 30/82 (36.6%) isolates showed no synergism with any inhibitors.

Considering these results, we evaluated the performance of a previously proposed modification of the disk synergy test to enhance the detection of MBL and Class A carbapenemases in dual producers, by using ceftazidime-avibactam/EDTA and aztreonam/boronic acid synergism, respectively. All isolates showed ceftazidime-avibactam/EDTA synergy, suggesting the presence of an MBL. KPC co-production inferred by a positive synergy with the aztreonam/boronic acid test was detected in 75/82 (91.5%) of the isolates. False negative aztreonam/boronic acid synergisms were associated with *K. pneumoniae* ST258.

## 4. Discussion

As the coronavirus disease pandemic swept through Argentina, most jurisdictions experienced peak hospitalizations at various times in 2020 (https://www.argentina.gob.ar/coronavirus/informes-diarios/sala-de-situacion/informes-especiales, accessed 17 March 2023). In this period, the country experienced an unprecedented increase in the rate of CPEs in a single year, matching the cumulative increase observed during the 9 years prior to COVID-19 (http://antimicrobianos.com.ar/ATB/wp-content/uploads/2022/11/Vigilancia-Nacional-de-la-Resistencia-a-los-Antimicrobianos-Red-WHONET-Argentina-Tendencia-2010-2021.pdf) (Accessed on 16 March 2023). The prevalence of Carbapenemase-producing *K. pneumoniae* in critical care areas showed the most dramatic increases, from 24% to 50% in blood and 26% to 42% in respiratory tract isolates (NRL personal communication). In addition, an increase in the burden of bacterial infections among inpatients during 2020 was documented [22]. According to the National Ministry of Health, a total of 115 patients per 10,000 discharges suffered an infection by a WHO priority pathogen, compared to 55/10,000 in 2019, with CRE experiencing the greatest growth, from 15.9 to 44.3 infections per 10,000 discharges (Dirección de Estadística e Información para la Salud, Ministerio de Salud de la Nación, personal communication). In this scenario of the spread of CPEs, strains with the co-production of KPC and MBL emerged for the first time in the country [12,13].

The epidemiology of resistant bacteria is under constant change, driven by dominant and successful clones. Argentina experienced the introduction and extensive dissemination of KPC-2-producing *K. pneumoniae* ST258 in 2010 [7]. In 2014, NDM-1 was first detected in *Providencia rettgeri* in Buenos Aires and disseminated in Enterobacterales species throughout the country [6]. Soon after, new changes occurred with the emergence of *K. pneumoniae* ST25 and ST307 detected from 2015 to 2017 in three hospitals in Buenos Aires City (CABA) [8]. That study also revealed the detection of *bla*_KPC-3_, which is rare in our country. Later on, a four-month outbreak in a hospital in CABA alerted the first detection of three *K. pneumoniae* ST307 co-producing KPC-3 and NDM-1 [23]. In the present study, we confirmed that ST307 has now outnumbered the former dominant clone ST258, suggesting the capability of hyper-epidemic ST307 to hold and disseminate two different carbapenemase genes in the hospital media. ST307 was initially reported as responsible for causing small outbreaks or sporadic cases in diverse regions of the world, producing either CTX-M-15 in Tunisia [24], KPC-2/3 in Texas [25], KPC-3 in Italy [26], and OXA-48 or the combination of *qnr* and CTX-M in Pakistan and Algeria [25,27]. Regarding South America, the first report of ST307 was from Colombia in 2013, causing an outbreak that extended to two hospitals and produced KPC-2 or KPC-3 [28]. ST307 was also reported from environmental samples in Brazil associated with CTX-M-15 [29]. It is interesting to note that ST307 has been mostly associated with the production of CTX-M (ESBL) or carbapenemases such as OXA-48 or KPC-2/3. Recently, a description of sporadic isolates of Enterobacterales co-producing KPC plus NDM was reported in our region [13]. However, and to the best of our knowledge, this is the first molecular characterization of clinical Enterobacterales isolates co-producing KPC plus an MBL involving several hospitals and cities. Molecular characterization revealed the dissemination ways of the dual carbapenemase producers, with intra-hospital outbreaks caused by one or more clones, as well as sporadic cases.

In Argentina, *bla*_NDM-1_ and *bla*_KPC-2_ used to be the most frequent carbapenemase alleles detected. Here, we found that in dual producers, there was a higher proportion of *bla*_NDM-5_ than *bla*_NDM-1_. We confirmed that *bla*_KPC-3_ remains an infrequent allele in the southern region, in this case, associated with *bla*_NDM-1_. Considering the global emergence of Enterobacterales co-producing two carbapenemases, a diverse combination of allelic variants were observed as NDM-1 plus OXA-232, KPC-2 plus VIM-1 [30], KPC-2 plus IMP-38-like [31], NDM-1 plus VIM-2 [32], VIM-1 plus OXA-48 [33], and KPC-2 plus NDM-5 [34]. Here, we also described diverse allelic combinations: KPC-2 plus NDM-5, KPC-2 plus NDM-1, KPC-3 plus NDM-1, KPC-2 plus IMP-8, and KPC-2 plus NDM-5 plus OXA-163. The coexistence of multiple resistance genes with similar hydrolytic activities in a single isolate could suggest a rapid evolution from acquiring and losing resistance genes in successful clones. The finding of diverse MBL genes (*bla*_NDM-5_ or *bla*_NDM-1_ or *bla*_IMP-8_) in a more uniform genetic background of strains harboring *bla*_KPC-2_ (e.g., CC307 and CC11), could suggest a more recent acquisition of MBL genes into well-established *K. pneumoniae* KPC-2-producing clones.

It was documented that aztreonam/avibactam may have a therapeutic advantage against MBL-producing strains, with additional resistance to aztreonam [35]; it showed a 60% reduction in the risk of mortality compared to other active antibiotics, significantly less clinical failure on day 14, and a shorter length of hospital stay [35]. In this sense, the American Society of Infectious Diseases recently recommended the use of ceftazidime/avibactam in combination with aztreonam as the preferred treatment option for NDM and other metallo-β-lactamases infections [36]. In this work, we showed that the aztreonam/avibactam combination was the most active antimicrobial against KPC plus MBL dual producers. This can be explained because avibactam, a diaza-bicyclo-octane (DBOs) molecule, inhibits Class A enzymes, such as KPC and other potential ESBL, Class C or D enzymes of hydrolyzing aztreonam, which is not hydrolyzed by Class B enzymes [37]. The same rationale for the aztreonam/avibactam combination could be extrapolated to other DBO combinations, such as relebactam. Relebactam is also a DBO inhibitor approved in combination with imipenem for the treatment of serine carbapenemase infections [37]. It shares the same inhibition profile as avibactam, except for Class D enzymes. To date, there is only one report comparing the in vitro activity of these two antimicrobial combinations against MBL- plus serine-β-lactamases co-produced in *K. pneumoniae* [38]. Maraki S. et al. reported no differences in the activity of aztreonam combined with either avibactam or relebactam, with synergism present in 97.5% of the strains [38]. Here, when we challenged aztreonam/relebactam with dual carbapenemase producers, unlike in the previous report, we observed ST-dependent activity, with ST258 being less inhibited by this combo. Further studies are required to understand the molecular mechanisms behind the largest MICs for aztreonam/relebactam in ST258, but we hypothesize additional aztreonam resistance mechanisms mediated by narrow or broad-spectrum Class D enzymes could be involved.

The accumulation of carbapenem-resistance mechanisms in clinical isolates represents a clear challenge for phenotypic tests. In this work, we observed that up to 80% of the strains had one or both carbapenemases undetected with the classic disk synergy test of EDTA and boronic acid disks placed close to carbapenem disks. In the same way, mCIM/eCIM mostly reported the presence of a single serine enzyme. Simple changes in disk approximation tests, using substrates affected by only one of the enzymes of the combination, could become a proficiency test for KPC and MBL phenotypic detection as described [39]. Briefly, aztreonam, which is intrinsically capable of evading Class B enzymes, could be a reporter of a Class A carbapenemase if susceptibility is restored by boronic acid. In addition, ceftazidime/avibactam resistance could become a subrogate marker of MBL if susceptibility is restored by EDTA. In this work, we demonstrated that this disk approach was able to reveal KPC and MBL co-production in 91% of the isolates. Misdetection was linked to ST258, probably due to additional resistance mechanisms against aztreonam (as mentioned above) which would be refractory to boronic acid inhibition. To extrapolate these phenotypic recommendations, it is vital to know the local circulation of at least the high-risk clones.

## 5. Conclusions

The dissemination of KPC plus MBL dual carbapenemase-producing isolates was driven in Argentina by the expansion of successful high-risk clones during the COVID-19 pandemic. The accumulation of carbapenem-resistance mechanisms in clinical isolates restricts treatment options. Consequently, the emergence and dissemination of double carbapenemase producers in Enterobacterales in high-risk clones stress the importance of a rapid microbiological diagnosis and the operative and articulated implementation of infection prevention and control programs.

## Figures and Tables

**Table 1 pathogens-12-00479-t001:** Distribution of clonal complexes (CC), sequence types (ST), pulsotypes (PFGE), hospitals (Htal) and carbapenemase combinations. * Include other K. pneumoniae STs and species.

Species	CC	ST	No. Isolates	PFGE	No. Isolates	Htal. Code (n)	Carbapenemase (n) *
*K. pneumoniae*	307	307	37	A	21	H1(1); H3(1); H17(1)	*bla*_KPC-2_ + *bla*_NDM-5_ (3)
						H23(1)	*bla*_KPC-2_ + *bla*_NDM-1_ (1)
						H1(4); H3(1); H17(10); H23(1)	*bla*_KPC_ + *bla*_NDM_ (16)
						H7(1)	*bla*_KPC_ + *bla*_IMP_ (1)
				C	3	H2(1)	*bla*_KPC-2_ + *bla*_NDM-1_ (1)
						H2(2)	*bla*_KPC_ + *bla*_NDM_ (2)
				F	2	H5(1)	*bla*_KPC-2_ + *bla*_NDM-1_ (1)
						H5(1)	*bla*_KPC_ + *bla*_NDM_ (1)
				Q	2	H6(1)	*bla*_KPC-2_ + *bla*_NDM-5_ (1)
						H6(1)	*bla*_KPC_ + *bla*_NDM_ (1)
				E	1	H2(1)	*bla*_KPC-2_ + *bla*_NDM-1_ (1)
				H	1	H12(1)	*bla*_KPC-2_ + *bla*_NDM-1_ (1)
				P	1	H27(1)	*bla*_KPC-2_ + *bla*_IMP-8_ (1)
				S	1	H1(1)	*bla*_KPC-2_ + *bla*_NDM-5_ (1)
				AA	1	H11(1)	*bla*_KPC-3_ + *bla*_NDM-1_ (1)
				AC	1	H20(1)	*bla*_KPC-2_ + *bla*_NDM-5_ (1)
				AD	1	H18(1)	*bla*_KPC-2_ + *bla*_NDM-5_ (1)
				AG	1	H16(1)	*bla*_KPC-2_ + *bla*_NDM-5_ (1)
				AH	1	H22(1)	*bla*_KPC-3_ + *bla*_NDM-1_ (1)
		5993	1	A	1	H1(1)	*bla*_KPC-2_ + *bla*_NDM-1_ (1)
	11	11	22	R	9	H1(9)	*bla*_KPC-2_ + *bla*_NDM-5_ (1)
							*bla*_KPC_ + *bla*_NDM_ (8)
				B	4	H2(4)	*bla*_KPC-2_ + *bla*_NDM-1_ (1)
							*bla*_KPC_ + *bla*_NDM_ (3)
				T	2	H1(2)	*bla*_KPC-2_ + *bla*_NDM-1_ (1)
							*bla*_KPC_ + *bla*_NDM_ (1)
				U	2	H24(1); H26(1)	*bla*_KPC-2_ + *bla*_NDM-5_ (1)
							*bla*_KPC_ + *bla*_NDM_ (1)
				J	1	H9(1)	*bla*_KPC-2_ + *bla*_NDM-5_ (1)
				O	1	H1(1)	*bla*_KPC-2_ + *bla*_NDM-5_ (1)
				W	1	H13(1)	*bla*_KPC-2_ + *bla*_NDM-5_ (1)
				AI	1	H1(1)	*bla*_KPC-2_ + *bla*_NDM-5_ (1)
				AJ	1	H25(1)	*bla*_KPC-2_ + *bla*_NDM-5_ (1)
		258	7	V	2	H8(2)	*bla*_KPC-2_ + *bla*_NDM-5_ (1)
							*bla*_KPC_ + *bla*_NDM_ (1)
				G	1	H4(1)	*bla*_KPC-2_ + *bla*_NDM-5_ (1)
				L	1	H2(1)	*bla*_KPC-2_ + *bla*_NDM-1_ (1)
				M	1	H10(1)	*bla*_KPC-2_ + *bla*_NDM-1_ (1)
				N	1	H7(1)	*bla*_KPC-2_ + *bla*_IMP-8_ (1)
				AB	1	H19(1)	*bla*_KPC-2_ + *bla*_NDM-5_ (1)
	45	45	3	D	3	H1(3)	*bla*_KPC-2_ + *bla*_NDM-1_ (1)
							*bla*_KPC_ + *bla*_NDM_ (2)
	147	147	1	AE	1	H18(1)	*bla*_KPC-2_ + *bla*_NDM-5_ + *bla*_OXA-163_ (1)
	219	219	1	K	1	H4(1)	*bla*_KPC-2_ + *bla*_NDM-5_ (1)
	485	485	1	I	1	H2(1)	*bla*_KPC-2_ + *bla*_NDM-1_ (1)
	15	5995	1	Z	1	H9(1)	*bla*_KPC-2_ + *bla*_NDM-5_ (1)
	ND	225	1	AF	1	H21(1)	*bla*_KPC-2_ + *bla*_NDM-5_ (1)
	ND	5994	1	Y	1	H15(1)	*bla*_KPC-2_ + *bla*_NDM-5_ (1)
	ND	2217	1	X	1	H14(1)	*bla*_KPC-2_ + *bla*_NDM-1_ (1)
*K. oxytoca*	NA	NA	2	1KO	1	H1(1)	*bla*_KPC-2_ + *bla*_IMP-8_ (1)
				2KO	1	H4(1)	*bla*_KPC-2_ + *bla*_NDM-5_ (1)
*C. freundii*	NA	NA	2	1CF	1	H13(1)	*bla*_KPC-2_ + *bla*_NDM-1_ (1)
				2CF	1	H28(1)	*bla*_KPC-2_ + *bla*_NDM-1_ (1)
*E. coli*	399	4774	1	NA	1	H14(1)	*bla*_KPC-2_ + *bla*_NDM-1_ (1)

**Table 2 pathogens-12-00479-t002:** Antimicrobial susceptibility to aztreonam/avibactam and aztreonam/relebactam of major *K. pneumoniae* clones.

	**Aztreonam-avibactam**								
	All sample	**Type of carbapenemase combinations**				** *K. pneumoniae* ** **clones**			
		KPC-2 + NDM-1	KPC-2 + NDM-5	KPC-2 + IMP-8	KPC-3 + NDM-1	ST307	ST11	ST258	Others *
	(n: 82)	(n: 16)	(n: 20)	(n: 4)	(n: 2)	(n: 38)	(n: 22)	(n: 7)	(n: 15)
MIC50 (µg/ml)	0.12	0.12	0.12	ND	ND	0.12	0.12	0.12	0.03
MIC90 (µg/ml)	0.25	0.25	0.25	ND	ND	0.12	0.25	0.25	0.5
Range (µg/ml)	<=0.03–0.5	0.03–0.25	0.03–0.5	0.03	0.03–0.12	<=0.03–0.25	0.12–0.25	0.06–0.25	0.03–0.5
	**Aztreonam-relebactam**								
	All sample	**Type of carbapenemase combinations**				** *K. pneumoniae* ** **clones**			
		KPC-2 + NDM-1	KPC-2 + NDM-5	KPC-2 + IMP-8	KPC-3 + NDM-1	ST307	ST11	ST258	Others *
	(n: 82)	(n: 16)	(n: 20)	(n: 4)	(n: 2)	(n: 38)	(n: 22)	(n: 7)	(n: 15)
MIC50 (µg/ml)	0.5	0.25	0.5	ND	ND	0.5	0.25	2	0.5
MIC90 (µg/ml)	2	1	2	ND	ND	1	0.5	8	1
Range (µg/ml)	<=0.03–8	0.03–1	0.03–8	0.03–0.12	0.25–4	0.06–2	0.12–0.5	0.5–8	0.03–2

* Include other *K. pneumoniae* STs and species.

## Data Availability

The data presented in this study are available in this article and in the Appendix A.

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
