# Peer review of "Emergence of Hyper-Epidemic Clones of Enterobacterales Clinical Isolates Co-Producing KPC and Metallo-Beta-Lactamases during the COVID-19 Pandemic"

_pathogens, 2023, doi:10.3390/pathogens12030479_

Round 1

Reviewer 1 Report

Table 2 contains information on the synergistic effect of aztreonam/avibactam and aztreonam/relebactam combinations on the main Klebsiella clones. However, it would be relevant to see this effect by type of carbapenemase produced, especially in the subgroups producing two carbapenemases. The authors could include another table with this information and the results could be commented in the discussion. 

Author Response

Q: Table 2 contains information on the synergistic effect of aztreonam/avibactam and aztreonam/relebactam combinations on the main Klebsiella clones. However, it would be relevant to see this effect by type of carbapenemase produced, especially in the subgroups producing two carbapenemases. The authors could include another table with this information and the results could be commented in the discussion.

A: Accepted. Table 2 was modified according to the recommendations of the reviewer. No significant differences was observed between the subgroups: KPC-2+NDM-1 and KPC-2+NDM-5. The other combinations, KPC-2+IMP-8 and KPC-3+NDM-1, were not analyzed because the number of isolates was too low. Finally, a sentence commenting these results was included in the revised version (lines 222-224).

Reviewer 2 Report

The authors describe the occurrence of hyperepidemic clones of Enterobacterales clinical isolates with a novel combination of genes encoding KPC and metallo-beta-lactamases from 28 hospitals. The article also describes their genetic relatedness and the genetic environment of the resistance genes. The article is providing insight into the occurrence of clinically relevant isolates which emerged, however, some points need to be addressed.

In the Methods (Abstract) you only mention where the isolates originated from, yet the conclusion describes which method was proven to be better for classification, yet this method is not mentioned in the Methodological part of the abstract. Furthermore, the entire sentence "Between March 2020 and September 2021, a sample of 82 Enterobacterales clinical isolates confirmed at the NRL as producers of a novel combination of blaKPC and a MBL gene were included. " would benefit from rephrasing and/or splitting as it seems to merge multiple ideas since blaKPC is a gene which can be carried/encoded, but not "produced" as you say.

Please find my questions below:

Why were only 82 representative isolates selected out of 118? Is there an equal number of isolates per location now?

Please rephrase "Plasmidic ESBLs" to "Plasmid-borne ESBLs" or similar.

In the case of the detection of plasmid-borne ESBLs, why weren't the present plasmids typed?

Author Response

We would like to thank all the suggestions made by reviewer 2. We have modified the manuscript accordingly in the revised version.

Q: In the Methods (Abstract) you only mention where the isolates originated from, yet the conclusion describes which method was proven to be better for classification, yet this method is not mentioned in the Methodological part of the abstract. Furthermore, the entire sentence "Between March 2020 and September 2021, a sample of 82 Enterobacterales clinical isolates confirmed at the NRL as producers of a novel combination of blaKPC and a MBL gene were included. " would benefit from rephrasing and/or splitting as it seems to merge multiple ideas since blaKPC is a gene which can be carried/encoded, but not "produced" as you say.

A: Accepted. The abstract was modified according to the recommendation.

Q: Why were only 82 representative isolates selected out of 118? Is there an equal number of isolates per location now?

A: Accepted. The selection of the isolates included all jurisdictions and institutions. In addition, we excluded those isolates suspected to be part of a hospital outbreak, or multiple isolates from a single patient. We have now improved the description of the isolate selection in the M&M section of the revised manuscript, as follow: “A sample of 82/118 representative isolates positive for blaKPC and an MBL gene were selected for further molecular study, contemplating one isolate per patient referred from all geographic regions, institutions, and years of isolation and excluding isolates that may be part of an outbreak.” Lines 85-89.

Q: Please rephrase "Plasmidic ESBLs" to "Plasmid-borne ESBLs" or similar.

A: Accepted. Line 203.

Q: In the case of the detection of plasmid-borne ESBLs, why weren't the present plasmids typed?

A: Partially accepted. We understand your point, however the main objective of the manuscript is to report and alert the emergence of multiple carbapenemase-producing Enterobacterales. Appropriate plasmid typing (association of bla gene with its specific plasmid Inc group) requires either long plus short read sequencing, which is too expensive for 82 isolates, or the complex combination of several methodologies like biparental conjugation, PCR-based replicon typing, S1-nuclease plus Southern-blot hybridization with specific bla and inc-group probes, among others. All of this, requires additional budget and at least one and a half extra wet-lab year which harms the urge to report this problematic emergence.